# Lindqvist@Nanoporous MOF-Based Catalyst for Effective Desulfurization of Fuels

**DOI:** 10.3390/nano12162887

**Published:** 2022-08-22

**Authors:** Simone Fernandes, Daniela Flores, Daniel Silva, Isabel Santos-Vieira, Fátima Mirante, Carlos M. Granadeiro, Salete S. Balula

**Affiliations:** 1LAQV/REQUIMTE & Department of Chemistry and Biochemistry, Faculty of Sciences, University of Porto, 4169-007 Porto, Portugal; 2CICECO—Aveiro Institute of Materials, Department of Chemistry, University of Aveiro, 3810-193 Aveiro, Portugal

**Keywords:** zeolitic imidazolate framework, Lindqvist compound, isopolyanion, desulfurization, oxidative catalysis, ionic liquid, hydrogen peroxide

## Abstract

An effective and sustainable oxidative desulfurization process for treating a multicomponent model fuel was successfully developed using as a heterogeneous catalyst a composite material containing as an active center the europium Lindqvist [Eu(W_5_O_18_)_2_]^9−^ (abbreviated as EuW_10_) encapsulated into the nanoporous ZIF-8 (zeolitic imidazolate framework) support. The EuW_10_@ZIF-8 composite was obtained through an impregnation procedure, and its successful preparation was confirmed by various characterization techniques (FT-IR, XRD, SEM/EDS, ICP-OES). The catalytic activity of the composite and the isolated EuW_10_ was evaluated in the desulfurization of a multicomponent model fuel containing dibenzothiophene derivatives (DBT, 4-MDBT and 4,6-DMDBT) with a total sulfur concentration of 1500 ppm. Oxidative desulfurization was performed using an ionic liquid as extraction solvent and aqueous hydrogen peroxide as oxidant. The catalytic results showed a remarkable desulfurization performance, with 99.5 and 94.7% sulfur removal in the first 180 min, for the homogeneous active center EuW_10_ and the heterogeneous EuW_10_@ZIF-8 catalysts, respectively. Furthermore, the stability of the nanocomposite catalyst was investigated by reusing and recycling processes. A superior retention of catalyst activity in consecutive desulfurization cycles was observed in the recycling studies when compared with the reusing experiments. Nevertheless, the nanostructure of ZIF-8 incorporating the active POM (polyoxometalate) was shown to be highly suitable for guaranteeing the absence of POM leaching, although structural modification was found for ZIF-8 after catalytic use that did not influenced catalytic performance.

## 1. Introduction

The combustion of transportation fuels has been one of the biggest global concerns in the last decades, due to environmental and health risks. In particular, sulfur compounds present in fuels can cause equipment corrosion, catalyst deactivation and, more importantly, the emission of sulfur oxides (SOx) [1,2]. The dangers associated with the emission of SOx, such as air pollution and acid rain, have led to the implementation of restrictive legislation limiting the sulfur content in commercialized transportation fuels to 10–15 ppm in Europe and the USA [3,4]. Hydrodesulfurization (HDS) is the conventional method used in refineries, because it is very effective in the removal of thiols, sulfides and dissulfides. However, it requires high temperatures (300–400 °C) and high hydrogen pressure to achieve the production of ultra-low-sulfur fuels [5,6]. The removal of certain sulfur compounds, such as benzothiophene derivatives, is more difficult to achieve by HDS, demanding even harsher conditions [7,8,9]. Therefore, alternative and/or complementary methods to current desulfurization technology have been developed. Oxidative desulfurization (ODS) is one of the most promising methods, reconciling sustainability with cost-efficiency when compared with HDS [8,10,11]. ODS operates efficiently under mild conditions, at atmospheric pressure, and without the need for hydrogen consumption [1,4,11]. Oxidative desulfurization occurs in two main steps: extraction and catalytic oxidation [2]. In the first step, a polar solvent is required in order to extract the sulfur compounds from the fuel [3,6,12,13,14]. Ionic liquids have demonstrated high desulfurization efficiency for extracting sulfur compounds from fuels and also for promoting highly effective catalytic oxidation [13,14,15,16,17,18,19]. However, the main key to achieving ultra-low-sulfur fuels is the adequate choice of catalyst and oxidant with which to oxidize the sulfur compounds into sulfoxides and sulfones [20,21,22,23,24]. Hydrogen peroxide has been widely used in ODS due to its low environmental impact, since the only resulting sub-product is water [13,14,25,26,27,28,29,30,31,32,33].

Polyoxometalates (POMs) are oxometallic anionic clusters that possess structural diversity, thermal stability and interesting redox properties [34,35,36]. POMs have been widely used as active catalytic centers in oxidative desulfurization [1,17,20,22,24,25,28,30,31,32,33,37]. The presence of lanthanide ions in the structure of POMs confers unique photoluminescent properties, and recently the application of lanthanide-containing POMs in ODS has been shown to be an effective strategy with very promising results [9,38,39]. However, the study of the catalytic potential of Lindqvist-type [Ln(W_5_O_18_)_2_]^n−^ (LnW_10_), where Ln = Eu^3+^, Gd^3+^, Tb^3+^), is still in its early stages [40,41,42,43].

The high solubility of POMs in polar solvents makes for difficult recovery and recycling of catalysts. The immobilization of active POM centers in solid porous supports, such as metal–organic frameworks (MOFs), allows the heterogenization of POMs and its reusability [44,45]. MOFs are hybrid materials obtained from the covalent connection between inorganic metal nodes using organic linkers [46]. ZIF-8 (zeolitic imidazolate framework) is a widely studied MOF with structural simplicity, ease of synthetic reproducibility, and potential for scale-up [47,48]. The existence of pores of adequate internal diameter (11.6 Å) and aperture (3.4 Å) allows the encapsulation of guests in the cavities while preventing leaching from the cavities [33,49,50].

In previous works from our research group, various Keggin-type POMs were incorporated into different porous MOFs to prepare active heterogeneous catalysts for oxidative desulfurization [1,30,33,39,46,51,52,53,54,55,56,57,58]. To the best of our knowledge, no reports can be found in the literature presenting Lindqvist-type isopolyanions incorporated into MOF supports for use as heterogeneous catalysts. The use of Lindqvist-type materials with a europium metal center supports the preparation of a more stable bridged-polyoxometalate, which incorporates a high number of tungsten octahedrons, promoting the activation of the oxidant, and thus increasing the desulfurization efficiency of the catalyst. This work reports the successful preparation of EuW_10_ isopolyanion encapsulated in an appropriate MOF structure (Figure 1), the Zeolitic Imidazolate Framework, ZIF-8, and the application of the novel composite EuW_10_@ZIF-8 to desulfurize a multicomponent fuel under sustainable conditions.

## 2. Experimental Section

### 2.1. Characterization Methods

Tetradecane (C_14_H_30_, >99.0%), zinc(II) nitrate hexahydrate (Zn(NO_3_)_2_·6H_2_O, 99.0%), sodium tungstate dihydrate (Na_2_WO_4_·2H_2_O, ≥99.5%), glacial acetic acid, 2-methylimidazole (2-MIM, 99.0%), europium chloride hexahydrate (EuCl_3_·6H_2_O, 99.9%), 1-butyl-3-methylimidazole hexafluorophosphate (C_8_H_15_F_6_N_2_P, >97%, [BMIM]PF_6_), hydrogen peroxide (H_2_O_2_, aq. 30%), dibenzothiophene (C_12_H_8_S, 98%, DBT) and 4-methyldibenzothiophene (C_13_H_10_S, 96%, MDBT) were obtained from Sigma-Aldrich. N,N-dimethylformamide (C_3_H_7_NO, 99.99%, DMF), methanol (MeOH, analytical reagent grade), ethanol (C_2_H_5_OH, >99.8%, EtOH) and decane (C_10_H_22_, ≥99.0%) were purchased from Fisher. Acetonitrile (CH_3_CN, >99.5%, MeCN), 1 and benzothiophene (C_8_H_6_S, >95%, BT) were acquired from Fluka. Potassium chloride (KCl, 99.5%) was obtained from Merck. 4,6-dimethyldibenzothiophene (C_14_H_12_S, 95%, DMDBT) were acquired from Acros Organic. None of these were subjected to further treatment or purification.

Fourier-transformed infrared (FT-IR) spectra were acquired using the attenuated total reflectance (ATR) operation mode of a PerkinElmer FT-IR System Spectrum BX spectrometer, and all the representations are shown in arbitrary units of transmittance. Elemental analysis for W was performed by ICP-OES on a Perkin-Elmer Otima 4300 DV instrument at the University of Santiago de Compostela, Spain. Powder X-ray diffraction (PXRD) patterns were obtained at room temperature on a Rigaku Geigerflex diffractometer operating with a Cu radiation source (*λ*_1_ = 1.540598 Å; *λ*_2_ = 1.544426 Å; *λ*_1_/*λ*_2_ = 0.500) and in a Bragg-Brentano θ/2θ configuration (45 kV, 40 mA) at the Analytical Laboratory of the Faculty of Sciences and Technology of the NOVA University Lisbon, Portugal. Intensity data were collected by a step-counting method (step 0.026°) in continuous mode in the 3 ≤ 2θ ≤ 50° range, and all representations are shown in arbitrary units of intensity. Scanning electron microscopy (SEM) and electron dispersive X-ray spectroscopy (EDS) analysis were performed at “Centro de Materiais da Universidade do Porto” (CEMUP, Porto, Portugal). The studies were performed in an FEI Quanta 400 FEG ESEM high-resolution scanning electron microscope equipped with an EDAX Genesis X4M spectrometer working at 15 kV. Samples were coated with an Au/Pd thin film by sputtering using SPI Module Sputter Coater equipment. Catalytic reactions were periodically monitored by GC-FID analysis carried out in a Bruker 430-GC-FID chromatograph. Hydrogen was used as carrier gas (55 cm·s^−1^) and fused silica Supelco capillary columns SPB-5 (30 m × 0.25 mm i.d.; 25 μm film thickness) were used. 

### 2.2. Materials Preparation

#### 2.2.1. Synthesis of Europium Polyoxometalate (EuW_10_)

The potassium salt of the europium Lindqvist derivative [Eu(W_5_O_18_)_2_]^9−^ was prepared through an adaptation of a previously reported method [59,60]. Na_2_WO_4_·2H_2_O (5.020 g; 15 mmol) was dissolved in 7 mL of water and the pH adjusted to 7 with glacial acetic acid. After heating the solution at 90 °C, a hot solution of EuCl_3_·6H_2_O (557 mg, 1.5 mmol) dissolved in H_2_O (2 mL) was added dropwise. The mixture was kept at 90 °C under stirring for 30 min, and then a previously prepared KCl solution (1.120 g, 15 mmol in 8 mL of H_2_O) was added. The solution was cooled in air conditions and kept in the fridge for 72 h. A white precipitate was collected, washed with EtOH and dried in a desiccator over silica gel. TGA showed a mass loss of 4.2% up to 120 °C, corresponding to the loss of seven H_2_O hydration molecules (Appendix A), leading to the formula K_9_[EuW_10_O_36_]·7H_2_O.

#### 2.2.2. Preparation of the Composite Material EuW_10_@ZIF-8

The ZIF-8 was prepared following an experimental procedure adapted from previously reported methods [61]. An initial solution of Zn(NO_3_)_2_·6H_2_O (900 mg, 3 mmol) in MeOH (30 mL) was magnetically stirred for 15 min. Then, a solution of 2-methylimidazole (2-MIM, 995 mg, 12 mmol) in MeOH (10 mL) was slowly added. The resulting reactional mixture was magnetically stirred for 24 h at ambient temperature, after which the resulting material was isolated by centrifugation, washed three times with MeOH, and dried at 60 °C under vacuum for 16 h.

The preparation of the composite was performed through the immobilization of the EuW_10_ in the porous ZIF-8 support through an impregnation procedure for POM@MOF composites [52,53,62]. The procedure consisted of the preparation of an ethanolic solution of EuW_10_ (90.2 mg; 0.031 mmol), followed by the addition of previously dried ZIF-8 (150 mg) and stirring of the mixture for 96 h at room temperature. The product was recovered by centrifugation, washed with MeOH, and dried in a desiccator over silica gel.

ZIF-8: selected FT-IR (cm^−1^): 1574 (w), 1458 (s), 1421 (s), 1385 (s), 1308 (s), 1146 (s), 995 (s), 758 (s), 692 (m), 420 (s).

EuW_10_@ZIF-8: Anal. Found (%): W, 28.6; loading of EuW_10_: 0.156 mmol per g; selected FT-IR (cm^−1^): 1577 (w), 1458 (s), 1419 (s), 1306 (s), 1144 (s), 993 (m), 932 (w), 887 (m), 835 (m), 752 (s), 692 (m), 420 (m).

### 2.3. Oxidative Desulfurization 

Desulfurization studies were performed with a model fuel prepared with DBT, 4-MDBT and 4,6-MDBT, in decane (500 ppm of each sulfur compound). The reactions were carried out under air in a closed borosilicate vessel with a magnetic stirrer and immersed in a thermostatically controlled liquid paraffin bath at 70 °C. Catalytic oxidative reactions were performed in a biphasic system with a 1:1 model fuel/extraction solvent with the ionic liquid [BMIM]PF_6_. In a representative experiment, a certain amount of the heterogeneous catalyst equivalent to 3 µmol of active POM (EuW_10_) was added to [BMIM]PF_6_. After adding the model fuel, the system was stirred for 10 min at 70 °C. The oxidative catalytic step was then initiated with the addition of aqueous H_2_O_2_ 30% (75 μL) to the reaction mixture. Tetradecane was used as a standard in the periodical monitorization of the sulfur content by GC analysis. 

## 3. Results and Discussion

### 3.1. Materials Characterization

The resulting EuW_10_@ZIF-8 composite material was characterized using several techniques, including Fourier transform infrared spectroscopy (FT-IR), powder XRD, thermogravimetric analysis (TGA), inductively coupled plasma–optical emission spectrometry (ICP-OES) and scanning electron microscopy (SEM) coupled with energy dispersive X-ray spectroscopy (EDS). The FT-IR spectrum of EuW_10_@ZIF-8 (Figure 1) exhibits the typical bands arising from the ZIF-8 framework, namely the bands associated with the stretching modes of the C=N and C–N bonds located 1577 and 1144 cm^−1^, respectively. The bands located at 993 and 752 cm^−1^ are ascribed to the bending modes of C–N and C–H bonds, respectively, while the band at 692 cm^−1^ arises from the out-of-plane bending of the ring. The band located at 420 cm^−1^ is assigned to the stretching vibration of the Zn–N confirming the coordination between the metal center and organic ligand [63,64]. The presence of additional bands can be observed in the spectrum of the composite when compared with the spectrum of ZIF-8. These bands are located at 937 cm^−1^, which can be assigned to the terminal ν_as_(W–O_d_) stretching mode, as well as at 887 and 835 cm^−1^, ascribed to the corner-sharing ν(W–O_b_–W) stretching modes of EuW_10_, respectively [60,65,66,67]. Moreover, the presence of the europium POM in the composite material was further confirmed by elemental analysis (ICP-OES). The results reveal a tungsten amount of 28.6 wt%, leading to a EuW_10_ loading of 0.156 mmol per g, which confirms the successful immobilization of the POM onto the porous ZIF-8 structure.

The crystalline structures of EuW_10_@ZIF-8 composite and the solid support were investigated by powder XRD. Figure 2 exhibits the patterns of both materials in the 5 < 2θ < 50° range. The results reveal highly crystalline materials with nearly identical diffraction patterns composed by the characteristic diffraction peaks of the ZIF-8 crystalline structure. The main diffraction peaks located at 2θ = 7.4, 10.5, 12.8, 14.9, 16.6, 18.2, 24.7 and 26.8° can be indexed to the (011), (002), (112), (022), (013), (222), (233) and (134) reflections. The high similarity between the diffraction patterns points out to the stability of the MOF by retaining its crystalline structure after the EuW_10_ incorporation process. Although an additional low-intensity peak can be detected at 2θ = 8.7°, the main diffraction peaks of EuW_10_ are absent from the EuW_10_@ZIF-8 pattern, suggesting its homogenous distribution within the MOF framework [68,69].

The morphology and chemical composition of the MOF support and the EuW_10_@ZIF-8 composite material were studied by SEM/EDS techniques. The SEM image of ZIF-8 (Figure 3A) reveals uniform particles with a morphology of rhombic dodecahedrons, which is in good agreement with the reported literature [70,71,72]. SEM images were used to estimate the average particle size (ImageJ v1.53e; https://imagej.nih.gov/ij/; accessed on 5 February 2022; 50 measurements), leading to a mean particle size of 554 ± 20 nm. The EDS spectrum of ZIF-8 (Appendix A) shows the presence of zinc as the main constituent as well as nitrogen from the organic ligand (2-methylimidazole). The SEM micrograph obtained for the EuW_10_@ZIF-8 composite material (Figure 3B) also exhibits rhombic dodecahedron-shaped particles with an average particle size of 567 ± 16 nm, suggesting that the MOF structure remained intact in the final composite. The EDS spectrum (Appendix A) shows, besides the Zn and N elements from the ZIF-8 support, the presence of W which further confirms the successful immobilization of EuW_10_ in the composite material.

The thermal stabilities of the support and the EuW_10_@ZIF-8 composite material were evaluated by thermogravimetric analysis (TGA) up to 1000 °C. In the TGA obtained for ZIF-8 (Appendix A), it is possible to observe a mass loss (11.8%) in the range 150–400 °C, which is most likely due to the evaporation of solvent molecules of methanol from inside the pores. Nevertheless, the most pronounced mass loss (69.7%) starts at ca. 550 °C and can be attributed to the decomposition of the organic ligand (2-methylimidazole), thus resulting in the collapse of the structure [73].

The TGA of EuW_10_@ZIF-8 (Appendix A) reveals a superior thermal stability of the composite when compared with the ZIF-8 support. In fact, the first mass loss (4.4%) is only detected at approximately 390 °C which should correspond to the evaporation of adsorbed solvent molecules. Afterwards, the main mass loss (51.8%) occurs at ca. 590 °C exhibiting a similar profile to the step observed in the ZIF-8 support and can be assigned to the degradation of the organic ligand together with the EuW_10_ anions. 

### 3.2. Catalytic Desulfurization Studies

The catalytic performance of the composite EuW_10_@ZIF-8 was investigated for the desulfurization of a multicomponent model fuel prepared with the most representative refractory sulfur compounds in real diesel, namely dibenzothiophene (DBT), 4-methyldibenzothiophene (4-MDBT) and 4,6-dimethyldibenzo-thiophene (4,6-DMDBT). Catalytic studies were performed at 70 °C in a biphasic liquid–liquid system based on equal volumes of the extraction solvent 1-butyl-3-methylimidazole hexafluorophosphate ([BMIM]PF_6_) and the model fuel. Aqueous hydrogen peroxide was used as oxidizing agent. The desulfurization process occurred in two main steps: extraction and catalytic oxidation. During the first 10 min and before the addition of oxidant, an initial extraction occurs to transfer certain amount of the sulfur compounds from fuel to the [BMIM]PF_6_ phase. The initial extraction equilibrium is reached, and the second step is initiated by adding hydrogen peroxide oxidant to the system. The metal-catalyzed oxidation of the extracted sulfur compounds is initiated (i.e., the oxidation of sulfur compounds to the corresponding sulfoxides and/or sulfones presented in the extraction solvent). The sulfur oxidation promotes a continued sulfur extraction from the diesel phase by decreasing the concentration of non-oxidized sulfur compounds in the extractant phase since no oxidation products were detected in the model diesel phase. The mechanism involved in the oxidation of benzothiophene and dibenzothiophene derivative compounds involved an initial formation of peroxo-POM active species via nucleophilic attack of the oxidant on the W^VI^ atoms of EuW_10_. Oxidation of sulfur compounds into sulfoxides occurs by transfer of an oxygen atom from the intermediate active species, which are then regenerated into the initial W^VI^ species. Subsequent nucleophilic attack on sulfoxides leads to the formation of the corresponding sulfones [9,25,27,41,43]. The desulfurization efficiency was determined by the periodical analysis of the model fuel phase by gas chromatography. Pristine ZIF-8 was tested as catalyst showing no activity in the studied reaction. Initially, the catalytic activity of the EuW_10_@ZIF-8 composite was compared with the homogeneous EuW_10_ (Figure 4). The desulfurization profile reveals a considerable extraction of sulfur compounds during the initial extraction step (50.2% and 68.3% of desulfurization for the heterogeneous EuW_10_@ZIF-8 and the homogeneous EuW_10_, respectively). Both catalysts were also demonstrated to be highly efficient for the desulfurization of the sulfur multicomponent model fuel. After 3 h of the process, near-complete desulfurization was achieved using the EuW_10_@ZIF-8 composite (94.7%). The homogeneous EuW_10_ was shown to be slightly more efficient than the composite, since after only 1 h, a desulfurization of 98.9% was achieved. The individual desulfurization for each sulfur compound in the presence of EuW_10_@ZIF-8 is depicted in Figure 5. The difficulty to remove the sulfur compounds from the model fuel increases according to the following order: DBT > 4-MDBT > 4,6-DMDBT, which is in good agreement with previously reported studies [4,74,75]. This tendency is correlated with the steric hindrance introduced by the methyl groups that hinders the nucleophilic attack, delaying the oxidation of the sulfur compounds to sulfoxides and/or sulfones. In fact, the removal of DBT from the model fuel was greatly promoted, with complete desulfurization being achieved after only 2 h. As for 4-MDBT, nearly total desulfurization (97.2%) was achieved after three hours of reaction, while only 87.9% was achieved for 4,6-DMDBT. Despite the superior catalytic activity of the homogeneous catalyst, a major drawback is its inability to be removed from the reactional medium and recycled in consecutive catalytic cycles. 

The recycling capacity of the solid EuW_10_@ZIF-8 catalyst and the reuse ability of the EuW_10_@ZIF-8/[BMIM]PF_6_ system were evaluated by performing five consecutive desulfurization cycles. The reuse method consisted of the preservation of the composite and the ionic liquid-phase EuW_10_@ZIF-8/[BMIM]PF_6_ during five consecutive cycles. The sustainability of this procedure is higher, since the same portion of extraction solvent and catalyst were used in these consecutive cycles. After each cycle, the model fuel was removed from the system and a new desulfurization cycle was performed by adding a novel portion of sulfurized model fuel as well the H_2_O_2_ oxidant, maintaining all the experimental conditions. In the case of the recycling procedure, the solid catalyst was recovered, washed with water and acetonitrile, dried, weighed, and used again in a new desulfurization cycle while maintaining the same experimental conditions. Figure 6 displays the results obtained after 4 h for the reusing and recycling processes. It is noteworthy that the recycling process of EuW_10_@ZIF_8_ demonstrates the retention of the desulfurization efficiency during the various cycles, with near complete desulfurization after 4 h. On the other hand, the reuse process of the EuW_10_@ZIF_8_/[BMIM]PF_6_ system presented a decrease in catalytic efficiency after the first cycle. Between the 2nd and the 4th cycles, the reuse process caused an accumulation of oxidized products (mainly sulfones) in the [BMIM]PF_6_ extraction phase. This caused a decrease in sulfur transfer from the fuel to the ionic liquid phase, and consequently a decrease in the desulfurization efficiency. This behavior has been attributed to the accumulation and consequent saturation of the extraction phase during the reuse process with the oxidized sulfur compounds (sulfones), which prevents further transfer of sulfur compounds from the non-polar phase (model diesel) to the ionic liquid phase, decreasing the desulfurization efficiency of the system [27,76]. After the 4th cycle, the solid catalyst was recovered, washed, and used in a new desulfurization cycle. This promoted an increase in desulfurization efficiency, which was observed for the 5th cycle of the reuse process. Therefore, the results show that the saturation of the extraction phase prevents the reutilization of the solid catalyst for longer desulfurization cycles, since the oxidative reaction should occur in the extraction phase (sulfones and/or sulfoxides were observed in the model fuel phase), and a clean medium promotes a higher diffusion of reactants to the catalytic active center located at the ZIF-8 framework.

The mechanism involved in the oxidation of benzothiophene and dibenzothiophene derivative compounds catalyzed by POMs using H_2_O_2_ as oxidant has been well described in the literature [9,52,75,77]. The initial formation of peroxo-POM active species occurs by nucleophilic attack of the oxidant on the W^VI^ atoms of EuW_10_. Oxidation of sulfur compounds into sulfoxides occurs by transfer of an oxygen atom from the intermediate active species, which are then regenerated into the initial W^VI^ species. Subsequent nucleophilic attack on sulfoxides lead to the formation of the corresponding sulfones.

### 3.3. Comparison with Other Catalysts

Recently, our research group reported the desulfurization of a multicomponent model fuel using homogeneous Lindqvist POMs (including EuW_10_) and compared their activities with Keggin-type POM (Table 1) [43]. The superior efficiency of the Lindqvist-type structure was observed, requiring a smaller oxidant dosage to reach complete desulfurization. In the present work, the heterogenization of the active Lindqvist EuW_10_ was strategically envisaged through its encapsulation into the framework of ZIF-8. The selection of this MOF as a host for the POM anion was due to the match between POM dimensions (near 10 Å) and the ZIF-8 pore size (11.6 Å), with a small pore window (3.4 Å) [78]. The small pore window prevents POM leaching, and at the same time can promote the diffusion of reactants towards the EuW_10_ catalytic active center. Scarce work can be found in the literature reporting POMs@ZIF-8 catalysts for oxidative desulfurization, and the majority only reports their application in single-component DBT model fuel [50,79]. The only work presenting the application of a POM@ZIF-8 catalyst using a multicomponent model fuel was reported by our research group, and involved the incorporation of an imidazolium-based Keggin-type polyoxomolybdate ([BMIM]_3_PMo_12_O_40_) into a ZIF-8 framework ([BMIM]_3_PMo_12_O_40_@ZIF-8) [33]. Under identical experimental conditions, near-complete desulfurization was achieved after 1 h; however, a structural alteration was observed after the first use of the catalyst. For this reason, the stability of the EuW_10_@ZIF-8 composite was investigated by using it in consecutive desulfurization cycles.

### 3.4. Catalyst Material Stability

The stability of the heterogeneous catalyst was evaluated by recovering the composite after catalytic use (EuW_10_@ZIF-8-ac). The catalyst was separated by centrifugation, washed with water and acetonitrile, and dried in a desiccator over silica gel. The powder XRD characterization of EuW_10_@ZIF-8-ac reveals that the composite undergoes structural transformation after catalytic use (Appendix A). In fact, the ZIF-8 is known to undergo thermally and pressure-induced phase transformations as previously reported by our research group and others [33,80]. In this case, the main diffraction peaks in the powder XRD pattern of EuW_10_@ZIF-8-ac suggest the structural transformation of ZIF-8 into the ZIF-L polymorph as a consequence of the applied experimental conditions [81,82]. The morphology and chemical composition of EuW_10_@ZIF-8-ac were assessed using SEM/EDS techniques (Appendix A). The SEM images showed a higher degree of aggregation between the particles, which was most likely due to remaining solvent with high viscosity in the recovered material, as suggested by the presence of P and F from [BMIM]PF_6_ in the EDS spectrum. The morphology of the particles also seems to have changed after catalytic use, most likely as a consequence of the previously discussed phase transformation. Nevertheless, the EDS spectrum of EuW_10_@ZIF-8-ac still exhibits the presence of Zn, N and W, with similar relative intensities to the spectrum of the as-prepared composite. This result, together with the retention of the high desulfurization results along the recycling studies suggests the minimal occurrence or even absence of EuW_10_ leaching.

## 4. Conclusions

The heterogenization of the active Lindqvist isopolyoxometalate EuW_10_ was performed in this work following an impregnation procedure to prepare the EuW_10_@ZIF-8 nanocomposite. The oxidative catalytic performance of this nanocomposite was compared with the homogeneous Lindqvist [Eu(W_5_O_18_)_2_]^9−^ (abbreviated as EuW_10_) for the desulfurization of a multicomponent model fuel. Only a slightly superior catalytic efficiency of the homogeneous was observed during the first 2 h of the process and practically complete desulfurization was achieved after 3 h. The reutilization and recycling processes of the solid catalyst were investigated over five consecutive desulfurization cycles. Due to the high saturation of the [BMIM]PF_6_ extraction phase after the first cycle, a decrease in desulfurization efficiency was detected. However, the activity was maintained for a number of consecutive cycles if the solid catalyst was recycled by separation and solvent washing was performed between cycles. Stability was further confirmed by different characterization techniques. The nanoporous ZIF-8 was shown to be highly suitable for promoting the high catalytic activity of the Lindqvist active center.

## Data Availability

Not applicable.

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
