# Peer review of "Lindqvist@Nanoporous MOF-Based Catalyst for Effective Desulfurization of Fuels"

_nanomaterials, 2022, doi:10.3390/nano12162887_

Round 1

Reviewer 1 Report

Recommendation: Publish after major revisions noted.

Comments:
Balula and co-authors present a study on the Lindqvist@Nanoporous MOF-based catalyst for effective desulfurization of fuels. Detailed experimental and characterizations were conducted. Overall, the manuscript is well organized. Thus I recommend it to be published in nanomaterials after addressing the following concerns:

1. As the author demonstrated in Fig. 6, “After the 4th cycle the solid catalyst was recovered, washed and used in a new desulfurization cycle. This promoted an increase of desulfurization efficiency observed for the 5th cycle of the reusing process. Actually, the total desulfurization rate is just 78% even though after regeneration. Therefore, intrinsic cause inducing this decrease of desulfurization efficiency should be analyzed in the revised manuscript.

2. The oxidative desulfurization mechanism over this EuW10@ZIF-8 should be clearly clarified in the revised manuscript.

3. As is well-known that the oxidant-hydrogen peroxide has highly corrosive, high costs for storage and severe fire safety concerns. Moreover, the by-product generated by the hydrogen peroxide for the current state-of-art catalysts significantly enhances the resistance of mass transfer and thereafter is extremely unbeneficial for the ODS reaction. In sharp contrast, molecular oxygen present in the ambient ambient atmosphere has emerged as a competitive alternative to hydrogen peroxide owing to its abundance, low cost, safety and fewer side-reactions. Thus, the desulfurization efficiency over this catalyst using H2O2 as oxidant should be provided.

4. The performance comparison with the current state-of-the-art catalysts should be provided.

5. There are many format errors present in the main article, such as Reference 35, and so on. It should be carefully corrected throughout the whole manuscript.

Reviewer 2 Report

In this work, the heterogenization of the active Lindqvist isopolyoxometalate EuW10 was performed following an impregnation procedure to prepare the EuW10@ZIF-8 nanocomposite, which was confirmed by various characterization techniques (FT-IR, XRD, SEM/EDS, ICP-OES). The oxidative catalytic performance of this nanocomposite was compared with the homogeneous Lindqvist [Eu(W5O18)2]9− (abbreviated as EuW10) for the desulfurization of a multicomponent model fuel. The catalytic results showed a remarkable desulfurization performance with 99.5 and 94.7% of sulfur removal in the first 180 min, for the homogeneous active center EuW10 and the heterogeneous EuW10@ZIF-8 catalysts, respectively. Despite the superior catalytic activity of the homogeneous catalyst, a major drawback is its inability of being removed from reactional medium and recycled in consecutive catalytic cycles. A superior retention of catalyst activity in consecutive desulfurization cycles was observed in the recycling studies when compared with the reusing experiments. Overall, the work done in this article is relatively novel and comprehensive. It could be considered for publication after some revisions.

1.     Please define POM and ZIF-8 on their first occurrence in the abstract.

2.     Please check "One of the most promising method is the oxidative desulfurization (ODS) which conciliates sustainability and cost-efficiency when compared with HDS, since ODS operates efficiently under mild conditions, atmospheric pressure and without the need of hydrogen consumption" for grammatical errors in the introduction.

3.     Please indicate the meaning of "rt" in scheme 1 for easy understanding by readers.

4.     It is recommended that "methods" in heading 2.1 be designated as characterization methods.

5.     In Section 3.1 it is suggested to delete "Porous zinc-imidazolate ZIF-8 MOF was selected as solid support for the immobilization of the europium decatungstate [Eu(W5O18)2]9- anion (abbreviated as EuW10). The immobilization was performed through an impregnation method using an ethanolic solution of the potassium salt of EuW10 following a previously reported procedure by our group for the preparation of POM@MOFs composite materials.” because the material preparation procedure has been mentioned before. It is unnecessary to repeat it here.

6.     Please check whether "vibrational spectroscopy" in "The resulting EuW10@ZIF-8 composite material was characterized by several techniques, including vibrational spectroscopy (FT-IR)" in Section 3.1 should be changed to "Fourier Transform infrared spectroscopy".

7.     It is recommended that the corresponding energy bands of the FT-IR map be clearly marked in Figure 1.

8.     Please indicate the third paragraph in section 3.1, where is "Image J"。

9.     Please check the description "The EDS spectrum of ZIF-8 (Fig. S1A) shows the presence of zinc as the main constituent as well as nitrogen from the organic ligand (2-methylimidazole)" for errors because the C content is shown in Fig. S1 also very high.

10.   Please check "The desulfurization profile reveals a considerable extraction of sulfur compounds during the initial extraction step (50.2 % and 68.3 % of desulfurization for the homogeneous EuW10 and the heterogeneous EuW10@ZIF-8, respectively)" in Section 3.2 for errors, because this contradicts the following statement.

11.   I don't understand what the second paragraph of Section 3.2 means here, whether to consider adjusting it to other sections.

Reviewer 3 Report

An article by Fernandes et al is devoted to a novel catalyst for oxidative fuel desulfurization based on europium polyoxotungstate incorporated into a Zn-based MOF matrix. The article is written in good language and contains interesting research.
The main question unclear to the reviewer is the choice of europium as a metal. Did the authors plan to study the course of the reaction or poisoning of the catalyst from luminescent data? The radius of the europium compound, unlike other lanthanides, how does the key with the lock match the pores of the MOF? The participation of europium(II) compounds in the catalysis cycle is incredible. Europium is a relatively expensive lanthanide, and could be successfully replaced by cheap elements at the beginning of the Ln-series. At least some justification for the choice of metal is necessary.
Otherwise, I have no significant comments. Figure 1 can either be labeled for the bands of the individual oscillations mentioned in the text, or removed from text to the SI. In the "Materials and Methods" section, the subscript is incorrectly indicated in the formula of sodium tungstate.
I think that the work can be published in Nanomaterials, but the rationale for the choice of metal in the text should be added.

Round 2

Reviewer 1 Report

The authors have already resolved the concerns from reviewers. It could be accepted as it is.

Reviewer 2 Report

This paper can be accepted now.